# Reading Performances of Illness Scripts, Clinical Authority, and Narrative Self-Care in Samuel Beckett's *Malone Dies* and Jérôme Lambert's *Chambre Simple*

**Swati Joshi [1],\*** and **Claire Jeantils [2],\***

[1] Humanities and Social Sciences Department, Indian Institute of Technology Gandhinagar, Gandhinagar 382355, India

[2] Langue et Littérature Française, THALIM, Sorbonne Nouvelle, 750002 Paris, France

\* Correspondence: swati1992.joshi@gmail.com (S.J.); claire.jeantils@sorbonne-nouvelle.fr (C.J.)

**Abstract:** *Malone Dies* (1956) by Samuel Beckett and *Chambre simple* (2018) by Jérôme Lambert present the narratives of precarity in the clinical setting, wherein the clinical caregivers view the suffering of the patients as a spectacle and chart out pre(script)ions and pro(script)ions for them. Both novels open on a note of uncertainty. This paper examines the narratives of fear and anxiety of the institutionalized patients (probably) in the mental asylum in *Malone Dies* and the public hospital in *Chambre simple*. The caregivers in both novels represent the voice of medical authority who focus on cure rather than care, providing their patients food and medications or conducting tests. Hence, Malone and le Patient are compelled to develop artistic coping mechanisms of self-care, reclaiming the ownership of the self. In *Malone Dies*, the abatement of in-person care and the fear of spending time in isolation before death motivates Malone to devise the narratives. Malone is the sole performer and spectator of his performance of patienthood. Similarly, le Patient chooses the position of the spectator, thus turning upside down the "spectacle" of the epilepsy script, where the patient is viewed as the performer of catharsis by the clinical audience. Here, the lens of performance studies helps us understand clinical caregivers' emphasis on preparing an illness script that governs Malone and le Patient's script of narrative self-care. We argue that caregivers' expectations pressurize patients with chronic conditions to implement forms of artistic self-care in clinical settings.

**Keywords:** performance of patienthood; illness script; artistic self-care; medical institution

## 1. Introduction

> You're asked to be the perfect patient, who's ennobled by her illness and makes everyone else around her feel better . . . . (Westenfeld 2022)

Meghan O'Rourke, in Adrienne Westenfeld's interview, highlights that clinical care implies the obedient performance of clinical expectations. She describes that "medicine is always evaluating patients," (Westenfeld 2022) as it analyses the good patient script and its performance, despite the patient's traumatic confrontation with painful symptoms and risks of the treatment. Traditionally, doctors study the patient's behavior in what the medical humanities call an illness script to make "specialized knowledge structures that link clinically relevant information about general disease categories" (Lubarsky et al. 2015, p. e63). Here, the clinical audience employs "clinical gaze as a perceptual act sustained by a logic of operations" to observe the patients' corporeal and non-corporeal performances (Foucault [1973] 2003, p. 133). The Foucauldian understanding of the gaze corroborates the lens of performance studies to analyze the clinical caregivers' intention behind formulating the illness script in Samuel Beckett's *Malone Dies*[1] (Beckett [1959] 1994]) and Jérôme Lambert's *Chambre simple* (2018).

Through the analysis of the aforementioned texts, we argue about the role of medical humanities to question this clinical care approach that documents the performance of

the patient script as the illness script[2] in the case of histories incorporating the patients' physiological, psychological, and emotional deviance, cooperation, etc., without lending them the space to exercise their agency. It is expressed through narrative self-care, which is a way of taking care of oneself by incorporating narrative practices into one's daily life. In this paper, we understand it broadly, encompassing written texts, daydreaming, idiolect, etc. Both novels showcase how the care recipients are compelled to engage in narrative self-care when their respective caregivers either abandon or choose to cure them rather than care for them. Here, the caregivers equate the performance of cure with the performance of care. We discuss how the performance of patienthood necessitates the caregivers' performance.

## 2. Performing Patienthood

The performances[3] of the patienthood by Malone and le Patient, in Samuel Beckett's *Malone Dies* and Jérôme Lambert's *Chambre simple*, respectively, take place (probably) in a mental asylum and a public hospital. Against the backdrop of clinical gaze and uncertainty, the patients question their arrival and the duration of confinement in a clinical setup. The French version of *Malone Dies* makes the similarity of the experience of Malone and le Patient of being trapped in their respective precarious circumstances even more intimate. Malone expresses his perplexity hinting at his loss of memory, "Je ne me rappelle pas comment j'y suis arrivé. Dans une ambulance peut-être, un véhicule quelconque certainement"[4] (Beckett 1951, p. 14). While Malone meditates on how he got to his "chambre semble être moi" (Beckett 1951, p. 13), translated as ("room seems to be mine" (Beckett [1959] 1994, p. 183)), le Patient ponders over his consciousness and wonders if he was conscious enough to witness how he was brought to his "chambre simple": "J'ai repris conscience sur le brancard d'une ambulance ou d'une voiture de pompiers, je ne sais plus" (Lambert 2018, p. 11) ("I regained consciousness on the stretcher of the ambulance or the fire brigade's car, I don't know."). It is evident from the afore-cited excerpts that both the ailing protagonists are bewildered about their whereabouts and how they reached there.

Beckett's *Malone Dies* is the second installment of the trilogy—*Molloy* (1955), *Malone Dies* (1956), and *The Unnamable* (1959)—wherein the narrator does not know how he arrived in a residential clinical care setup. *Malone Dies* appears as a continuation of *Molloy*, wherein the narrator's constant emphasis lies on uncertainty. Molloy begins his narrative with, "I do not remember how I got here. In an ambulance perhaps, a vehicle of some kind certainly" (Beckett [1959] 1994, p. 9) and so does Malone when he says, "I do not remember how I got here. In an ambulance perhaps, a vehicle of some kind certainly" (Beckett [1959] 1994). Perhaps Molloy, who thinks he could not have arrived alone in an asylum-like institute becomes an alone Malone in the second part of the trilogy. It is a monologic novel, as we do not hear the responses of Malone's caregivers.

*Chambre simple* is a polyphonic novel set in a hospital, wherein each character is allowed to voice their opinion on the story, but the narration is about and centered around one character called 'the Patient' who has been admitted to the hospital because of an epileptic seizure. His name is known to the reader only by the end of the novel. He has been seizure-free for quite some time and did not expect to experience all of that again any time soon. The characters who gravitate around him are his lover Roman, his caregivers Ellia and Maxime, and another patient, Marco. Each chapter gives a more detailed image of epilepsy, patienthood, and patient care.

Hence, Malone and le Patient construct their conjectures. Le Patient thinks, "Peut-être qu'un neurone n'a pas fait son boulot" ("Perhaps a neuron didn't do its job") (Lambert 2018, p. 12). Malone surmises, "I have often amused myself with trying to invent them, those same lost events . . . perhaps I was stunned with a blow, on the head, in a forest perhaps, yes now that I speak of a forest I vaguely remember a forest" (Beckett [1959] 1994, pp. 183–84). Malone and le Patient's efforts to reconstruct possible conditions that were responsible for landing them in the clinical care milieu showcase their endeavors to claim ownership of their suffering and their

attempts at narrative self-care. Neither remember the past events, and they are compelled to accept their present prescribed clinical entrapment.

It is important to note that, while the former presents the criticality of the suffering via loss of memory, the latter presents to the readers the patient's concerns of being conscious enough to witness his performance of patienthood and his caregiver(s)' performance of care.

Malone's performance of the patient script aligns with le Patient, who vaguely remembers being brought by the ambulance or some vehicle to the hospital. "J'ai repris conscience sur le brancard d'une ambulance ou d'une voiture de pompiers, je ne sais plus" (Lambert 2018, p. 11) ("I regained consciousness on the stretcher of the ambulance or the fire brigade's car, I don't know"). Their realization of patienthood comes from the clinical gaze that questions them (in the case of le Patient) or that monitors the schedule of their meals and disposal of waste (in the case of Malone).

The Foucauldian interpretation of the clinical gaze presents it as the language of medicine "that did not owe its truth to speech but to gaze alone" (Foucault [1973] 2003, p. 85). Medicine, according to Foucault, considers visual language appropriate to learn about the disease. Furthermore, he expounds that the clinical gaze has the paradoxical ability to hear a language as soon as it perceives a spectacle, quite contrary to the ethics of care that demand the patient to be "comforted, (and) not displayed" (Foucault [1973] 2003, p. 102). The twentieth century takes a turn in understanding the clinical gaze, with Rita Charon's suggestion of ameliorating the narrative competencies of the caregivers to humanize clinical encounters via empathy (Charon 2006). From the 1960s to the present, researchers of the humanities have criticized clinical spaces for their lack of care. Custers (2015) highlights the journey of the sophistication of illness scripts, catalogs its major components, and elaborates on its role in accurate diagnosis. The research on illness scripts might ostensibly seem patient-oriented, as it is the patient's narrative. Nevertheless, the process of the formulation of the illness script is caregiver-centric, facilitating them to decide on its structure, content, and language. Our analysis of the selected texts challenges the literary representation of the patient as a passive storyteller and demands the inclusion of the patient as an active collaborator in the context of clinical care. We argue that fear and anxiety motivate Malone and le Patient, respectively, for performing a good patient script in the clinical space that does not facilitate patient agency.

We argue that, when a cure is not accessible, patients tend to care for themselves with artistic coping mechanisms, thereby reclaiming their agency and the ownership of the self. Against the backdrop of the difference between cure and care, and the capability of the patient to devise artistic self-care[5], our paper poses the following questions: (1) What are the structural loopholes while caring for institutionalized chronically ill patients? (2) When and why does an institutionalized patient experience lack of care? (3) Can literature teach us anything about clinical self-care? If yes, how?

The analysis of the aspects of the artistic mode of self-care, motivated by fear and anxiety due to the lack of in-person care in the clinical spaces, sits at the intersection of the Beckett studies and Medical Humanities. This contributes to the rich wealth of research conducted by Beckett scholars such as Laura Salisbury, Elizabeth Barry, Ulrika Maude, Jonathan Heron, and Rina Kim, among others who contributed to the "Beckett, Medicine, and the Brain" issue of Journal of Medical Humanities.[6] Moreover, the analysis of *Chambre simple* employing this perspective informs about epileptologic care, which, so far, has mainly focused on medication. Before studying the approach of the clinical caregivers, we describe the patients' experiences in their respective clinical care spaces.

As stated before, the patient's experience of fear in the mental asylum-like institute runs through the veins of the first two installments of *The Trilogy*, namely *Molloy* and *Malone Dies*. Molloy keeps repeating, "I am full of fear, I have gone in fear all my life, in fear of blows" (Beckett [1959] 1994, p. 22). This fear is carried forward in *Malone Dies*, as Malone fears how he would spend his time taking care of himself before his death. Malone's opening avowal, "I shall soon be quite dead in spite of all" (Beckett [1959] 1994, p. 179) echoes his fear of deviance as a result of the distant care of the caregivers. This motivates

him to perform the good patient script while waiting for a probable cure, possible care, and certainty of death. "I must be on my guard against throes ... quietly crying and laughing, without working myself into a state" (Beckett [1959] 1994, p. 180). Malone's narratives showcase how imperative it is for him to perform the good patient script, lest he be "abandoned" by his caregivers only to end up "in the dark, without anything to play with" (Beckett [1959] 1994, p. 181). To cope with this, he seems to have designed the following script for the performance of self-care of writing stories: "Present state, three stories, inventory, there. An occasional interlude is to be feared. A full programme" (Beckett [1959] 1994, pp. 182–83). Malone's experience of abandonment and lack of care can be best understood with Goffman's explanation of the interaction between two individuals that define a particular situation, even if one of the individuals is already obeying the passive role assigned to them. " ... when he appears before others, we must also see that the others, however passive their role may seem to be, will themselves effectively project a definition of the situation by virtue of their response to the individual and by virtue of any lines of action they initiate to him" (Goffman 1956, p. 3). Just as Malone's caregivers perceive him as a performer of the deviant script, he responds by writing stories as a mode of self-care. It facilitates the readers to learn about Malone's clinical caregivers from his perspective. The readers do not know what his caregivers think of him. The novel gives the impression of reading Malone's private journal that records his fears, his observations, his stories of self-care, and much more. This is also mentioned in *The Making of Samuel Beckett's Malone meurt/Malone Dies* (Van Hulle and Verhulst 2017), "In H. Porter Abbot's book *Diary Fiction, Malone Dies* is one of the examples of such a 'diary novel'" (Abbot 1984, p. 24). The manner of Malone's communication might appear to the readers similar to that of the series of interior monologues.

On the other hand, Lambert's novel uses the vocabulary of theatre performance to approach the superficiality of the clinical setting, which we investigated further. Le Patient experiences the contradiction as a comedian would; an individuality ("I", "je") is asked to play a role, without a precise identity ("l'Allongé" as in "the supine patient"), as he enacts a concept. Nobody wants to see his true personhood but the epileptic performance. Ellia and Maxime, the two caregivers of Le Patient, the assistant nurse and nurse, respectively, are aware of these mechanisms and blame the physicians and their working conditions for not having enough time to care for patients. They are forbidden to experience transference. Maxime repeats to himself: "Pas de transfert: je connais la règle d'or". ("No transference: I know the golden rule") (Lambert 2018, p. 24) and a colleague reminds Ellia when she is thinking about Le Patient: "Pas de transfert, hein" ("No transference, right") (Lambert 2018, p. 72). Hence, the epileptic character's identity is linked to his confinement to the bed and his passivity for receiving the cure. Thus, the enactment of care appears purely mechanical:

> Le Médecin-coryphée s'avance solennellement jusqu'au pied du lit, face au Patient. Il tient entre ses mains sa partition sur laquelle tout est écrit, tout est joué, tout est perdu. Ses yeux se penchent sur ses notes, regard vide, puis il se redresse lentement vers l'Allongé.
>
> Le temps de la Sentence approche (Lambert 2018, pp. 85–86).
>
> (The doctor-coryphaeus advances solemnly to the foot of the bed, facing the Patient. He holds in his hands the score on which everything is written, everything is played, everything is lost. His eyes bend over his notes, his gaze empty, then he slowly straightens up towards the Patient. The time of the Sentence approaches)[7].

Here, we see how impersonal these roles are. Even though interchangeable, they still feel as fixed as an antique play. There is a clear *mise en abyme* of theatricality. The text is not particularly specific to epilepsy; we all recognize this traditional atmosphere in a hospital. However, what is interesting is how the theatricality is threaded in the novel, among the different voices, to announce the epileptic fit itself. Here, by telling some stage directions that are usually hidden, the character shows the mechanisms behind this play. He reclaims

the situation by parodying it, thereby distancing the experience of the hospital. The lexical scope becomes juridic with the word "sentence". Theatricality and the codes of the classical drama enable us to analyze the central line "tout est écrit, tout est joué, tout est perdu" (all is written, all is played, all is lost) as an alexandrine meter that has the role of the sentence. The anaphora[8] and the rhythm underline the dramatic aspect of the medical diagnosis to parody it. This judges the patient's behavior and not his disease.

Once the stage directions are announced in *Chambre simple*, and the script of patient-hood written in *Malone Dies*, the characters appear on the scene. The performance of patienthood in *Malone Dies* also involves the discussion of the witness. While Malone's behavior as a patient is monitored by his caregiving spectators, his stories do not have an audience or a witness to testify to the catharsis of the pain. Malone, being the sole inventor, performer, and spectator of the self-prescribed narrative self-care, claims interactive clinical care. His ward is the stage, wherein the hand(s) of his caregiver(s) present(s) the props of care, and while Malone obeys his caregivers, he also audits the caregivers' performance of care. Hence, in both novels, the clinical ward becomes the site of collision of cares, wherein the patients and the clinical caregivers assume the dual roles of acting and viewing to demand a voice in the dialogue of care.

Malone's props of self-care (the pencil, paper, and the stick) help him in purging his fear to perform the good patient script. Malone's impromptu performance of a good patient's script by writing stories gives rise to a care script[9] that demonstrates his experience of lack of care. There is no one to listen to his stories. However, as he creates these stories, he reflects on the aspects of depiction and analyzes how the portrayal of each character in his story is the manifestation of his desires and fears.

The cogitation on catharsis makes room for the discussion of the theatre tradition to examine if le Patient is acting. To seize or not to seize? That is the question for him! The theatrical motif implies whether le Patient has any agency in the hospital.

> si j'étais capable de ne pas en souffrir, je pouvais—qui sait—choisir mes moments de crise (Lambert 2018, p. 47).

> even if I was not able to suffer from it, I could—who knows—choose my moments of seizure.

Thus, it is the seizure script that makes le Patient epileptic. The chronicity of the disease is lost in the process of hospitalization and is reduced to the symptom of a seizure. The number of seizures before and after hospitalization disappears in the performance of the good patient script. This stigmatizes le Patient and impacts the quality of care administered by the clinical caregivers.

## 3. Scripting Performances, Performing Scripts, and Narrative Self-Care

Script theory "proposes an explanation for how information is stored in and retrieved from the human mind to influence individuals' interpretation of events in the world" (Lubarsky et al. 2015, p. e62). This concept has been used in medical education to describe how medical knowledge is stored in the medical student's mind, organized in networks, under the label of "illness script" (Lubarsky et al. 2015, p. e66). The illness script allows the physician to see patterns and irregularities in symptoms. It generates a model ready to analyze a situation. We understand how it can objectify patients who, according to this theory, are either inside or outside a "normal" pattern of symptoms and emotional reactions.

We noticed an increase in interest in the medical education literature for the notion and teaching of illness script. That is why it is most relevant to question its applications. We wonder what has become of the bedside teaching technique, which has facilitated the birth and development of the clinic. Bedside teaching was reintroduced in the 1980s in medical training to promote interactions between medical students and patients. Illness script and bedside teaching share many similarities. They both develop an art of reading the patient and appear to be dangerous when not trainee-specific, disease-specific, and, most importantly, patient-specific (Garout et al. 2016).

(Foucault [1973] 2003) explains that bedside teaching views the sick body as a place of neutrality, whereon the doctor inscribes care in a specific way. Here, the physician's gaze translates the body into an obedient apparatus for the authoritarian alchemy of care. However, the difference between the illness script and bedside teaching is that bedside teaching is about what is immediate and not mediated. The student should come to the bedside with no preconceptions, whereas an illness script expects the patient to perform the assumptions of the clinical caregivers. In *Chambre simple*, le Patient describes what might happen to him based on his experiential knowledge of the disease. He is able to anticipate the illness script the doctor is going to see in his case because he has experienced it multiple times (See Lambert 2018, pp. 13–14). In both techniques, the clinical caregivers appear to be the spectators of pain. The texts highlight how caregivers' practices could be improved.

The clinical gaze disciplines the chronically ill patient to perform the good patient script to be eligible to receive clinical care:

> J'avais été jusque-là un patient exemplaire, un soigné docile et obéissant. Oui, je comprends, d'accord, je vais faire ce que vous dites ... J'ai moi aussi appris ma partition, reprenant le rôle de celui qui occupait mon lit avant, j'ai rejoint la grande farce des soins, la tragédie de la maladie, comptez sur moi (Lambert 2018, pp. 141–42).

> (So far I had been an exemplary patient, a docile and obedient patient. Yes, I understand, okay, I'll do as you say ... I too have learned my part, taking on the role of the one who occupied my bed before, I have joined the great farce of care, the tragedy of illness, count me in).

The epileptic patient knows his part perfectly and is forced to make sense of daily life without seizures. Because this is how the hospital seems to define a good patient, it is one who does not trigger seizures. Thus, for the illness script to work, there is necessarily a good patient script that comes with it. Le Patient should stay calm and not be disturbed by noises, lights, and lots of activities. He should also have "good hygiene"[10] and not put himself at risk. Le Patient acts this way because of the caregivers' deontic tone. The epilepsy script is reduced by the caregivers to a seizure script. Epilepsy implies seizures; thus, their focus should be avoiding seizures, and they seem to be trained only to react to that. The other aspects of the chronicity of the disease (headaches, memory loss, behavioral changes, anxiety, etc.) are not mentioned. This situation is quite common for people with epilepsy, as explained by previous authors (Pineau-Valencienne 2000).

However, not having seizures at all during his hospitalization seems as though he does not need to be taken care of, as if he is not ill. It is a mascarade where one can be a good patient if one acts as though a cure is in sight, whereas it will most likely never be achievable. Canguilhem underlines the relationship between doctors and the cure:

> "Ce qui les intéresse, c'est de diagnostiquer et de guérir. Guérir c'est en principe ramener à la norme une fonction ou un organisme qui s'en sont écartés." (Canguilhem [1966] 2018, p. 98).

> They are interested in diagnosing and healing. To cure is, in principle, to bring back to the norm a function or an organism that has deviated from it.

> Diagnosis, illness script, and ultimately cures are the holy trinity of occidental care. They act in a way to normalize a body that took a wrong path, made a mistake, or proved abnormal to the social and medical gazes.

In this novel, the epilepsy script is not presented as the educational tool western medicine might use. Rather, it seems to be a force that overshadows the whole novel and gives the reader the feeling that caregivers control le Patient's performance of the patient script.

Our paper asks why does le Patient listen to the caregivers if he is aware of the artificiality of the situation and their inability to care for him? Le Patient lacks medical knowledge and, thus, acts out of anxiety, which threatens his seizure-free time at the

hospital. Hence, the case history becomes the clinical source that incorporates le Patient's patienthood. Though le Patient's performance of the patient script shapes his case history, he is not its writer. The clinicians claim the authority to decipher and decide the ideal cure that suits le Patient.

> Tant que je ne réponds pas, j'ai le pouvoir. Une fois que la réponse passera de leur côté, ils décideront, je n'aurai plus voix au chapitre, je serai passif, je ne serai plus rien (Lambert 2018, p. 13).

> As long as I don't answer, I have the power. Once the answer is on their side, they will decide, I would no longer have a word to say, I will be passive, I will be nothing.

Here, medical knowledge appears to be the currency of exchange in the care relationship. It is also a vicious circle of hierarchy for le Patient and the other caregivers. Le Patient's nurse, who is a cut below the doctors in the hierarchy of caregiving, explains, "On accompagne mais on ne sert à rien tant qu'on ne sait rien. Ils gardent la connaissance de leur côté" (Lambert 2018, p. 31) ("We accompany but we are useless as long as we don't know anything. They keep the knowledge on their side"). The performance of the illness script is the source of knowledge for the clinical caregivers. Here, the readers can witness the violence of artificiality of the clinical setting that pressurizes and compels le Patient to suffer from the seizure, thereby justifying the hospitalization and performing the epileptic patient's role.

The pressure placed on the performance of the good patient script now segues into an examination of the artistic strategy of self-care employed by Malone while being confined in the clinical care setting. Malone suggests that, while waiting to receive care from his caregivers, he shall perform the script of self-care by telling himself stories that will be "calm" and "almost lifeless like the teller" (p. 180). These calm stories are possibly the creative mechanism of self-care devised by Malone in the absence of the embodied care catered by his caregivers at his institution. Here, Hermann's (2017) explanation of writing as a mode of narrative self-care is crucial:

> Writing is, at root, an externalizing act. When we write, we bring what is inside us to the outside; we put words, however indirectly or metaphorically or imperfectly, to what's inside of us, feelings and experiences that previously were not concrete. Language *is* the realization of thought—-it is how thought comes to be in the world, and it is the way that one recognizes it (p. 215).

*Malone Dies* acquaints readers with the need for Malone to be creative, which is understood better with Hermann's (2017) explanation of the role of creative writing in one's life. It facilitates the catharsis of emotions via narratives. This argument could be corroborated by Samuel Beckett's reflections on how writing helped him "to vent the pent" (Beckett [1959] 1994, p. 4). Catharsis for Malone is documenting his stories on paper. Writing stories helps him perform the good patient script, thereby channelizing his fear as a creative force. Hermann shares her personal view and experience of the dividend of the Narrative Medicine of finding "solace and freedom in the act of making things up—-changing my own experience so that it looks and *is experienced* other than the way it was for me, which then allows me see ways that it could have been different" (p. 219). Hermann's explanation of her intention of writing fiction while witnessing the clinical conditions of her loved ones and suffering the loss is pertinent to the analysis of Malone's stories. Malone, too, modifies his stories even if they are grounded in his experience of being a patient. He changes the personality of one of the fictitious nurses that is modeled on his nurse at the asylum. This leads us to discuss if stories need to show the objective truth.

The stories of Malone give insight into his expectations of his caregivers. However, they may appear far-fetched and more sexualized to the readers. We are trying to understand Malone's narratives as sthe readers of the story that the approach upposedly that of a lonely patient who waits for holistic care but receives the bare minimum distant care from his caregivers. He writes the stories of the Lamberts, the Saposcats, and Macmann. His

stories manifest the precarity he experiences in life. "Christian name? I don't know . . . What friends? I don't know" (Beckett [1959] 1994, p. 187). However, it is not just the manifestation of his personal experience that makes Malone's stories fascinating. The first story about the Saposcats shows how the parents are compelling their son to become a surgeon, but he prefers to work on the farms of the Lamberts and take long walks. This story demonstrates Sapo's parents' fear of not receiving clinical care. Hence, they want their son to become a doctor even though he does not seem interested in pursuing medicine. For the present article, the story of the lonely Macmann staying at a caregiving institute appears pertinent, as this story facilitates Malone to express his concerns, feelings, and experience regarding his caregivers. At Macmann's caregiving institute, he is cared for by a nurse called Moll, just as Malone is possibly cared for by the hand of a woman.

> One day, much later, to judge by his appearance, Macmann came to again, once again, in a kind of asylum. At first he did not know it was one, being plunged within it, but he was told so as soon as he was in a condition to receive news. They said in substance, You are now in the House of Saint John of God, with the number one hundred and sixty-six . . . Take no thought for anything, it is we shall think and act for you, from now forward (Beckett [1959] 1994, p. 257).

Macmann's clinical care settings bear description, unlike Malone's. As mentioned before, the readers are not aware of the "The room, or cell" (Beckett [1959] 1994, p. 257) wherein Malone resides, but one can conjecture that Macmann's room could be possibly modeled on Malone's. Macmann, in his room, sees that his caregivers follow the protocols and expect him to do the same. Macmann sees "men and women dressed in white" giving "instructions regarding his duties and prerogatives" (Beckett [1959] 1994, p. 257). One of them is Moll- "a little old woman, immoderately ill-favoured of both face and body" with "revolting features" (Beckett [1959] 1994, p. 258). Macmann is glued to his bed as Malone was to his. In fact, Macmann's bed is not only the site of clinical care but also becomes the locus of passion wherein "Hairy Mac and Sucky Molly" (Beckett [1959] 1994, p. 263) despite being "completely impotent they finally succeeded in summoning to their aid all the resources of the skin, the mucus and the imagination" (Beckett [1959] 1994, p. 261). Moll, as his primary caregiver, brings him food and teaches him the basics of hygiene. She also explains to him the things he is permitted and/or prohibited to do at this asylum, and she is also the one who engages actively in a sexual relationship with her care-recipient. Before delving deeper into Malone's intentionality to limn the sexualized care relationship of Moll and Macmann, it is important to have a glance at his description of their discommoding coitus:

> This first phase, that of the bed, was characterized by the evolution of the relationship between Macmann and his keeper. . . . The spectacle was then offered of Macmann trying to bundle his sex into his partner's like a pillow into a pillowslip, folding it in two and stuffing it in with his fingers" (Beckett [1959] 1994, p. 261).

The sexualized representation of care in Malone's narrative does not necessarily reflect his desire to receive sexual care from his caregivers, But, as we read further about Macmann's revulsion with Moll's "lips in particular" (Beckett [1959] 1994, p. 264) and their struggle to assert power over each other during the sexual act, Malone's intentions become clearer. The sexual act, in Macmann's story, can be understood as a power struggle with his clinical caregivers, and the bed could be perceived as a space wherein he intends to claim dominance (at least in his stories). Malone's description of Moll's attitude toward Macmann despite her deteriorating health shows Malone's attitude toward his caregivers. On a superficial level, Malone's narrative may seem to demonstrate his desire to be cared for by a self-effacing and self-sacrificing nurse who does not seek care even when she is at her worst but continues to care for Macmann till her death. Malone's description of Moll turning away from him while being glued to her chair for hours and vomiting, only getting up to clean after gaining some strength, showcases his perception of his caregivers.

Malone's narrative shows how Macmann and Moll witness their mutual suffering. For Malone, the reality of caregivers is restricted to the gauntlet that he can see. He does not know what his caregiver(s) look like or experience. In giving the voice, smell, and body to Moll in his narrative, Malone animates the caregiver, giving her a personality. Following his throes, Malone's caregivers follow a routine of the administration of care wherein he can now see just a hand that changes bowls for him. His perception of distant care instills fear in his mind, which he channels to write narratives.

> Not only am I left here, but I am looked after! This is how it is done now. The door half opens, a hand puts a dish on the little table left there for that purpose, takes away the dish of the previous day, and the door closes again. . . . They have thought of everything (Beckett [1959] 1994, pp. 648–49).

Malone informs the readers of the story that the approach of his caregivers was different previously and he is not sure, "Why should have the powers changed their attitude towards me?" (p. 183). Though readers, similar to Malone, are not aware of the cause of the distant care, it could possibly be because of the previously mentioned throes. Malone mentions that, initially, a woman "came right into the room, bustled about, enquired about my needs, my wants" (p. 185). What Malone sees "now is the gaunt hand and part of the sleeve . . . perhaps now it is another's hand" (p. 186). At this juncture, writing stories, no matter how fictitious, seems the only means of Hermann's concept of narrative self-care at hand for him. Malone constructs narratives about two kinds of caregivers. Moll does assert authority but also caters to his needs, and Lemuel (who replaces Moll after her death) beats Malone, takes him and the other patients on a boat, and sets sail before killing a patient and another lady. These narratives reflect Malone's imagination of caregivers since he does not see his own. This calls for our analyses of writing fiction.

Hermann (2017) explains, "Writing fiction, then, and writing about my experiences as if they happened to someone else, allowed me to objectify my 'truth' and freed me to triangulate my experience, so that I was not gripping the reality of it quite so tightly" (p. 219). The advantageous aspect of writing stories rather than memoir, elaborates Hermann, is that one does not have to stress themselves for remembering every "true" thing that happened to them and regret that they "couldn't possibly do justice to the experience" (p. 221) by failing to mention the fragments of the objective reality. This explanation proves that the artistic narrative self-care mechanism does not necessarily need to be true to life. Malone's stories are meant for self-care, not for objective clinical case notes. The act of writing, for Malone, is his performance of a good patient script.

Here, the discussion of narrative medicine's principle of expressing one's vulnerability through their writing becomes pertinent. In *Chambre Simple*, we find narrative self-care on different levels. The first one is in the narrative structure of the novel. The text is polyphonic (each chapter brings a different voice), which enables readers to hear the voices of nurses, care assistants, le Patient's lover, and even another patient, all acting in the same scene, but they all seem to be drawn to le Patient as if he is the only voice, imagining the other ones. Le Patient in *Chambre Simple* uses personal tales and daydreaming as a self-care tool. It allows le Patient to escape mentally before being able to escape physically from the hospital. These narratives are embedded in sensory experiences, and they seem lively for le Patient:

> «Au réveil, je me suis dit que c'était plus un souvenir qu'un rêve. Les images n'avaient pas la consistance duveteuse des songes, les contours étaient nets comme une gravure et je me souvenais de la température de l'eau, de la sueur glissant sur mes côtes comme une huile» (Lambert 2018, p. 59).

> Waking up I told myself that this was more a memory than a dream. The pictures didn't have the fluffy consistency of dreams, outlines were sharp just like an engraving and I remembered the water temperature, the sweat sliding down my ribs like an oil.

Narrative practice is inherently linked to physical sensations such as sweat and heat and touch. According to him, it is because the hospital inhibits his life experience that his mind is drawn to daydreaming. His insights lead him to conclude that both his body and his mind need sharp sensations to feel alive again and to "delineate" his body, as the supine position and the hospital organization encourage passiveness over physical exercise. Indeed, physical activities tend to be prohibited for persons with epilepsy. Le Patient knows what he is not supposed to do; thus, he confides to the diegesis about his physical needs and wishes (such as cycling in Berlin, pp. 57–58). Once again, what le Patient needs physically and mentally is the exact opposite of what the hospital instructs him.

In *Chambre simple*, narrative self-care demonstrates nuanced diction. Le Patient develops his idiolect to reciprocate the medical jargon that gives him anxiety. The epileptic seizure is turned into a "myriad of nettles"("myriade d'orties"), an "iodine fizz" ("pétillement iodé"), and "rising waves of mimosas and nitrites" ("ces montées en vagues de mimosas et de nitrites") (Lambert 2018, p. 39). Here, the epileptic seizure is not shameful but diverse, it is not public but private[11]. It can neither be understood nor controlled by someone else. Through the different narrative self-care practices, le Patient reclaims the situation and the ownership of the self.

Beckett's *Malone Dies* portrays the ownership of the self through the fear that propels Malone's performance of the good patient script through writing narratives. Our understanding of Malone's ownership of his struggles emerges from Fifield's (2012) analysis of *Malone Dies*. "*Malone Dies* deviates from the habitual experience of pain when Malone appears to exercise control over his pains . . . " (p. 126). Fifield's explanation shows Malone's ownership of his struggles and his attempts at self-care by controlling it. Malone's narrative of Macmann shows Malone's desire for collaborative care and his demand for the right to perform his agency, which can be also understood with James Thompson's explanation about the aesthetic value of care: "Aesthetic value is located in-between people in moments of collaborative creation, conjoined effort and intimate exchange . . . " (p. 46). The lack of interaction between Malone and his caregivers does not permit us to probe the aesthetics of care but rather compels Malone to perform aesthetic care. "Aesthetics are therefore on my side, at least a certain kind of aesthetics" (Beckett [1959] 1994, p. 182). Malone's performance of the good patient script not only makes him eligible to receive care from his caregivers, but, as Thompson (2020) says, that cooperative/collaborative performance also reciprocates care for his audience-his clinical caregivers.

Similarly, in *Chambre simple*, the intersubjectivity is found in the theatricality, which we explained previously. The prevalence of the theatricality throughout the novel, explicitly or implicitly, gives the impression that the characters in the hospital take full responsibility for the role they have to play, explaining it in total transparency to the reader/spectator.

## 4. Conclusions

In this paper, while analyzing the performance of the good patient script that governs the illness script of the clinical caregivers in Samuel Beckett's *Malone Dies* and Jérôme Lambert's *Chambre simple*, the discussion of the site of the performance is also important. "A setting"—the asylum-like institute for Malone and the hospital for le Patient—becomes the stage "of their performance" (Goffman 1956, p. 13). In this space, they perform their identities in a prescribed way that makes them eligible to receive care. In the clinical setting, we find two groups: powerful and powerless. Hence, the usage of 'I' in *Malone Dies* and the noun 'le Patient' in *Chambre Simple* corroborate what Goffman explains as habituating oneself within the total institutions that coerce one to perceive themselves in a particular way.

This answered the first question we raised in this paper of the structural loopholes of the care catered to chronically ill patients at the institutions. When the patient is admitted for a long time, any deviance from the expected performance of the good patient script makes them ineligible to receive embodied care from the caregivers. Hence, the answer to the second question of the lack of care experienced by the patient is somehow answered in

the explanation of the first one. The sole answer to the second question is the performance of the good patient script, to channel the negative emotions or condition of mind as a creative force to devise the artistic mechanism of self-care.

We argue that *Chambre simple* and *Malone Dies* are similar in some ways. Both novels demonstrate how chronically ill patients can develop artistic coping mechanisms such as stories (Malone) and daydreaming, theatricalization, and remembrance (le Patient) to take care of themselves. The patients exercise their agency by employing their experiential knowledge, physical needs, yearnings, and anxieties and hope to suggest to their clinical caregivers strategies for caring for chronically ill patients for self-care in the clinical setup. Literary language and form offer the necessary plasticity to express the experience of chronic illness patienthood in all its confusing and twisted aspects. Our study calls for a more patient-oriented model of care that relies on narrative medicine and leaves room and facilitates agency for patient self-care.

**Author Contributions:** Conceptualization, S.J. and C.J.; methodology, S.J. and C.J.; software, not applicable.; validation, not applicable.; formal analysis, S.J. and C.J.; investigation, S.J. and C.J.; resources, S.J. and C.J.; data curation, S.J. and C.J.; writing—original draft preparation, S.J. and C.J.; writing—review and editing, S.J. and C.J.; visualization, S.J. and C.J.; supervision, S.J. and C.J.; project administration, not applicable.; funding acquisition, not applicable. All authors have read and agreed to the published version of the manuscript.

**Funding:** This research received no external funding.

**Institutional Review Board Statement:** Not applicable.

**Informed Consent Statement:** Not applicable.

**Data Availability Statement:** Not applicable.

**Conflicts of Interest:** The authors declare no conflict of interest.

## Notes

1   Please refer to "'Beckett on the Wards': medical humanities pedagogy and 'compassionate care'" by Heron et al. (2015). There has been copious research on Samuel Beckett's works employing the lens of medical humanities to study the disabilities, pain, and suffering of the characters but nothing so far has been written on care.

2   Please refer to Schneider (2014, p. 126) for understanding the way scripts are defined in different disciplines. Schneider explains that the scripts are "culture-dependent" and help members of various communities participating in the process of communication to "know what to do or say, when and how" to enable efficient communication among them. This paper takes inspiration from the aforementioned explanation of the scripts to discuss how scripts of patienthood, illness, and care govern the clinical communication between the caregivers and the care recipients in the clinical care settings in the selected novels. Moreover, for reading on the writing of scripts in the context of Beckett's works, please refer Chapter 6, "Writing Scripts" in Abbot's (1996) *Beckett Writing Beckett*.

3   Here, the definition of the performance of patienthood is derived from Goffman's definition of performance. Goffman (1956) explains, "all the activity of an individual which occurs during a period marked by his continuous presence before a particular set of observers and which has some influence on the observers" (p. 13). Drawing on Goffman's explanation, we show that the physiological enactment of the care-recipient's suffering effectuates the caregivers' performance of care.

4   The English translation of this sentence is taken from Beckett [1959]'s (Beckett [1959] 1994) own translation of the French version in the further discussion.

5   Swati Joshi's thesis devises a care script that draws on Amartya Sen's (1993) *The Quality of Life* and Martha Nussbaum's (2000) *Women and Human Development: The Capabilities Approach* that emphasize the individual's abilities of self-care. However, the writers don't incorporate the capability of being interdependent for mutual care.

6   Please refer to the entire special issue of "Beckett, Medicine, and the Brain" of *Journal of Medical Humanities* edited by Elizabeth Barry in June 2016. This edited issue was the output of "AHRC-funded project run jointly between Warwick (Elizabeth Barry), Birkbeck (Laura Salisbury) and Reading (Ulrika Maude), which aimed to produce collaboration between literary and theatrical scholars and clinicians and researchers in psychiatry and neuroscience, and use the work of Samuel Beckett to interrogate current concepts of mental disorder" (Barry et al. 2012). Moreover, please refer to Ulrika Maude's (2008).

7   All the translations in this paper from French to English are done by Claire Jeantils (Forthcoming).

8   The repetition of the same group of words at the beginning of a syntagm.

9    Swati Joshi (Forthcoming) has devised the idea of the care script in her thesis inspired by the capabilities approach of Amartya Sen (1993) and Martha Nussbaum's (2000) capabilities approach. Care script literally means the patterns of care acquired by the caregivers and the care-recipients that help them in caring for the ones in need or themselves.

10   Drawing on the epileptical knowledge Jeantils (Forthcoming) wants to highlight that "good hygiene" is usually required from patients with epilepsy to not trigger seizures, that entails good sleep, no caffeine, no alcohol, etc.

11   Jeantils (Forthcoming) gathered and analysed a corpus of epilepsy narratives and argued that this type of deictic vocabulary, which is patient and disease-specific, requires nuanced understanding of the doctors, caregivers, and readers. This discussion appears in Jeantils' thesis.

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
