# Peer review of "Reading Performances of Illness Scripts, Clinical Authority, and Narrative Self-Care in Samuel Beckett’s Malone Dies and Jérôme Lambert’s Chambre Simple"

_humanities, doi:10.3390/h11060140_

Round 1

Reviewer 1 Report

I write this with an acknowledgment that I cannot speak to Lambert's Chambre Simple. I can, however, provide commentary on the article as it pertains to Beckett's Malone Dies.

I find the analysis that Malone performs narratives and constructs scripts of illness for self-care and cure intriguing. The use of theoretical texts regarding self-care and narrative therapy are convincingly used in the article. With additional work, this article could be an important contribution in Beckett Studies.

To get to this stage, I have some recommendations.

I would recommend that there is an engagement with Beckett Studies to help develop the analysis of Malone Dies. The only Beckett scholar mentioned (and only dealt with briefly in the footnotes) is Swati Joshi. Her thesis is not listed in the bibliography so there is no way a reader can follow up if interested in learning more. There are other Beckett scholars working on Beckett and the medical humanities as well as scholars who have written on the purpose of Malone's writing. It is important to show a knowledge of the work done in Beckett Studies.

I'd also like to see a stronger engagement with what Malone actually writes. His tales are full of grotesque images of Moll and grotesque sexually imagery. How does this factor into your argument? Similarly, I don't get a strong sense of Chambre Simple. I don't know the work, but a stronger engagement with the text of the novel would certainly help.

Towards the end of the article, the author/s note that 'Beckett's Malone Dies' portrays the ownership of the self through the fear that propels Malone's performance of the good patient script through writing narratives.' I am not convinced that 'ownership of the self' is ever achieved. Maybe this is because there isn't enough of a nuanced discussion of the text. I'd like a little more development of this point to be convinced. 

Author Response

We thank the editor for inviting us to make the changes that will strengthen our paper. We have added the brief synopsis of both the novels and have explained the aspects that are analysed at length in the paper. So, this addresses the recommendation to engage with the novels. We have also cited the references that help us corroborate the argument of the protagonists' ownership of struggles and self-care. As per the reviewer's suggestions, we have described the content of Malone's stories in Malone Dies. 

We have tried to address all the suggestions given by the reviewer. Please inform us if there are any further changes to be made.

Reviewer 2 Report

I really enjoyed the topic of this fascinating paper, and I think that it does an excellent job of bridging the topics of medical humanities, literary analysis, and sociology. My main concern is that the paper tries to do slightly too much without ensuring that the reader knows both the terms and the case studies, and it would benefit from a much clearer definition of the terms and context at the top of the argument. A few brief sentences that would provide synopses for Malone Dies (which is the better explained book of the two) and Chambre simple would go a long way, as would quick definitions of narrative self care. In general, the argument moves well, but it is sidetracked by jargon that is dropped in and then not explored or connected (for example, "dramaturgy techniques" in line 123, which should actually be "dramaturgical techniques", is an entirely different mode of analysis that we don't need to know here). I also think the authors need to engage at least a little more with the idea of performing narratives, as that is so crucial to this important argument. It's in Goffman, and it's in the work of others who write about "cultural scripts" and the like. I think that would really strengthen this argument. The authors also have several (more than a few, but not too many) moments where the sentence structure might be technically correct, but is formulated in an awkward way. I recommend that a native English speaker go through to correct these - I noticed about ten instances of this in the whole paper. Despite the room for improvement, I think these fixes are going to be easily done, and I truly hope that these authors revise and resubmit! 

Author Response

Thank you for your invaluable comments. As per your advice, we've provided elaborate synopsis of Chambre Simple, and explained the jargons incorporated in our paper. We have also provided the definition of cultural scripts as you'd suggested. We have used the software of Grammarly to edit our paper this time, so hopefully there will be fewer or no grammatical errors. But just in case, if there are any, we'd appreciate if you them to our notice. Thank you for sparing your time to review our paper. 

Round 2

Reviewer 1 Report

This revision is much improved. However, there are still areas that need fine-tuning:

The story of Moll still doesn't take into account the description of her and the sexual grotesque. However, the authors have added more about Malone's script.

There are still references used in the body of the text that do not appear in the Bibliography. Abbott is not listed in the bibliography, nor is Beckett's Malone meurt. I urge the authors do go through and make sure all works are cited.

At least one author's name is misspelled: Pim Verhulst is one.

I was surprised that Laura Salisbury wasn't incorporated. 

I recommend removing the assertation that 'the lack of in-person care in the clinical spaces have not been studied within the Beckett studies and Medical Humanities.' It's unnecessary to make such a claim, and it may not be true. There may be others who are studying this aspect of Beckett's work, and the authors' may just not be aware of others doing so.

There is still some lingering British spelling and typos. 

Author Response

Dear Editor,

We thank you for sparing time to read our paper and give us your invaluable comments to strengthen the content of our paper. We have made all the necessary edits like elaborating the sexual relationship between Moll and Macmann and have highlighted the grotesque aspects of their relationship. We have also included the scholarship of Salisbury, Barry, and Maude in the fields of Beckett Studies and Medical Humanities. We have added the missing texts in Bibliography and edited the spelling of Pim Verhulst. We have also replaced the British spellings with the American. If there are any further edits, please let us know. Thank you once again for all your efforts, time, and comments. We sincerely appreciate your inputs.

Warm regards

Reviewer 2 Report

I want to commend the authors for these excellent revisions! I feel that the new introduction makes all the difference, and now the scope of the argument, the terms it takes up, and the pertinent details of both novels are laid out in a way that enables more readers to appreciate this fascinating argument. The parallelism of cultural scripts/health scripts, and performance scripts is striking here, and the interdisciplinary nature of this essay will be of use to people in medical humanities, psychology/sociology, and theatre and performance fields. 

Author Response

Dear Reviewer,

We would like to thank you for sparing time to read our article and suggesting us the comments that have helped us shape and strengthen our arguments better. Thank you once again for your invaluable inputs. We sincerely appreciate your efforts and contribution.

Warm regards